# Transgenic interleukin 11 expression causes cross-tissue fibro-inflammation and an inflammatory bowel phenotype in mice

Wei-Wen Lim[1,2], Benjamin Ng[1,2], Anissa Widjaja[2], Chen Xie[1], Liping Su[1], Nicole Ko[2], Sze-Yun Lim[1], Xiu-Yi Kwek[1], Stella Lim[2], Stuart Alexander Cook[1,2,3,4‡]*, Sebastian Schafer[1,2‡]*

**1** National Heart Research Institute Singapore, National Heart Centre Singapore, Singapore, Singapore, **2** Cardiovascular and Metabolic Disorders Program, Duke-National University of Singapore Medical School, Singapore, Singapore, **3** National Heart and Lung Institute, Imperial College London, London, England, United Kingdom, **4** MRC-London Institute of Medical Sciences, London, England, United Kingdom

‡ These authors jointly supervised this work.
* sebastian@duke-nus.edu.sg (SS); stuart.cook@singhealth.com (SAC)

**Data Availability Statement:** All relevant data are within the paper and its Supporting Information files.

## Abstract

Interleukin 11 (IL11) is a profibrotic cytokine, secreted by myofibroblasts and damaged epithelial cells. Smooth muscle cells (SMCs) also secrete IL11 under pathological conditions and express the IL11 receptor. Here we examined the effects of SMC-specific, conditional expression of murine IL11 in a transgenic mouse (*Il11*[SMC]). Within days of transgene activation, *Il11*[SMC] mice developed loose stools and progressive bleeding and rectal prolapse, which was associated with a 65% mortality by two weeks. The bowel of *Il11*[SMC] mice was inflamed, fibrotic and had a thickened wall, which was accompanied by activation of ERK and STAT3. In other organs, including the heart, lung, liver, kidney and skin there was a phenotypic spectrum of fibro-inflammation, together with consistent ERK activation. To investigate further the importance of stromal-derived IL11 in the inflammatory bowel phenotype we used a second model with fibroblast-specific expression of IL11, the *Il11*[Fib] mouse. This additional model largely phenocopied the *Il11*[SMC] bowel phenotype. These data show that IL11 secretion from the stromal niche is sufficient to drive inflammatory bowel disease in mice. Given that IL11 expression in colonic stromal cells predicts anti-TNF therapy failure in patients with ulcerative colitis or Crohn's disease, we suggest IL11 as a therapeutic target for inflammatory bowel disease.

## Introduction

Non-striated smooth muscle cells (SMCs) line the walls of hollow organs and the vasculature. In adults, SMCs are not terminally differentiated and their cellular phenotype remains plastic. A variety of extracellular cues such as humoral factors, mechanical or oxidative stress and cell-cell interactions can induce a spectrum of cellular states ranging from contractile SMCs to highly synthetic and proliferative SMCs [1]. Synthetic SMCs are associated with a wide variety

**Funding:** The research was supported by the National Medical Research Council (NMRC; https://www.nmrc.gov.sg/) Singapore STaR awards to S. A.C. (NMRC/STaR/0029/2017), the NMRC Central Grant to the NHCS, Goh Foundation, Tanoto Foundation and a grant from the Fondation Leducq. S.S. is supported by the Goh Foundation and Charles Toh Cardiovascular Fellowship and by the National Medical Research Council Young Individual Research Grant (MOH-OFYIRG18nov-0003). A.A.W. is supported by the NMRC YIRG (NMRC/OFYIRG/0053/2017). The funders had no role in study design, data collection and analysis, decision to publish, or preparation of the manuscript.

**Competing interests:** S.A.C. and S.S. are co-inventors of the patent applications 'Treatment of fibrosis' (WO/2017/103108). S.A.C., S.S., W.W.L. and B.N. are co-inventors of the patent application 'Treatment of SMC mediated disease' (WO/2019/073057). S.A.C. and S.S. are co-founders and shareholders of Enleofen Bio PTE LTD, a company (which S.A.C. is a director of) that develops anti-IL11 therapeutics. All other authors declare no competing interests.

of vascular pathologies such as atherosclerosis or hypertension [1] and other disorders such as asthma [2] and inflammatory bowel disease (IBD) [3]. Many fibro-inflammatory diseases have a component, or are defined by, SMC dysfunction. This is exemplified by systemic sclerosis, which presents with global organ fibrosis and specific vascular abnormalities [4] and is characterized by elevated transforming growth factor beta (TGFB) 2 and interleukin 11 (IL11) expression in dermal stromal cells [5, 6]. This co-occurrence of fibrosis and SMC dysfunction may in part be explained by molecular similarities of the fibrogenic fibroblast-to-myofibroblast conversion and the SMC contractile-to-synthetic phenotype switch. Both these cellular transitions are characterized by extracellular matrix (ECM) production, cell proliferation, invasion and migration. They can also be triggered by the same extracellular cues including TGFB family members [1, 7].

We recently identified IL11 as a critical driver of fibroblast activation in the cardiovascular system, liver and lung downstream of a variety of pro-fibrotic factors including TGFB1 [8–10]. In a study from 1999, IL11 was also found to be secreted by vascular SMCs (VSMCs) in response to pathogenic stimuli, including interleukin 1 alpha (IL1A), TGFB and tumor necrosis factor (TNF) [11]. Although IL11 is upregulated in systemic sclerosis [6], TNF-resistant ulcerative colitis [12, 13] and asthma [14] and despite SMCs being a source of IL11 [11], the effect of IL11 function in SMC biology has not been studied. To address this gap in our knowledge, we generated an inducible *Il11* transgenic mouse to overexpress mouse *Il11* in myosin heavy chain 11 (*Myh11*)-positive smooth muscle cells (*Il11*^SMC). Here we characterized key organs that may be affected by SMC pathobiology in *Il11*^SMC mice to better understand the role of SMC-derived IL11.

## Materials and methods

### Mouse models

This study was carried out in compliance with the recommendations in the *Guidelines on the Care and Use of Animals for Scientific Purposes* of the *National Advisory Committee for Laboratory Animal Research* (NACLAR). All experimental procedures were approved (2014/SHS/0925) and conducted in accordance to the SingHealth Institutional Animal Care and Use Committee (IACUC). All mice were from a C57BL/6JN genetic background and were bred and housed in individually vented cages in the same room under ABSL-1 conditions in the SingHealth Experimental Medicine Centre and provided normal chow (Specialty Feeds) and water *ad libitum*. All research staff involved in animal studies underwent the *Responsible Care and Use of Laboratory Animal Course* (RCULAC, Singapore) prior to study commencement. Animals were euthanized at endpoint by ketamine (100 mg/kg) and xylazine (10 mg/kg) given IP, followed by the removal of vital organs and tissues.

Mice were scruffed to restrict motion during tamoxifen administration IP and monitored daily for clinical signs of distress and body weights were measured thrice per week upon tamoxifen induction. When rectal inflammation/bleeding was observed, the wound was gently disinfected with 70% methylated spirits and 10% povidone-iodine. Mice that displayed behavioral abnormalities, weight loss, and/or rectal bleeding were therapeutically treated with buprenorphine (0.1 mg/kg SQ) and enrofloxacin (5 mg/kg SQ) where necessary. Animals that did not recover with treatment or presented with deteriorated symptoms including pronounced weight loss (>20% over 1 week or >10% over 24 hours) and the development of rectal prolapse were euthanized following consultation with a veterinarian prior to the study endpoint and were treated as deaths.

**Smooth muscle-specific Il11 transgenic model.** To direct transgene expression in smooth muscle cells, we crossed the heterozygous *Rosa26-Il11* (Gt(ROSA)26Sor^tm1(CAG-Il11)Cook)

mouse [8] to the hemizygous *SMMHC-CreERT2* (B6.FVB-Tg(Myh11-cre/ERT2)1Soff/J) mouse [15] available from the Jackson Laboratory (031928 and 019079 respectively) to generate double heterozygous *SMMHC-CreERT2:Rosa26-IL11* offspring (referred to here as *Il11*^SMC mice). Only male *Il11*^SMC mice were utilized as the *Myh11-Cre/ERT2* transgene is inserted on the Y chromosome. To induce Cre-mediated *Il11* transgene induction, six week old *Il11*^SMC mice were intraperitoneal injected with 3 doses of 50 mg kg$^{-1}$ tamoxifen (tam; T5648, Sigma Aldrich) or an equivalent volume of corn oil vehicle (veh; C8267, Sigma Aldrich) for a week. Single hemizygous *SMMHC-CreERT2* littermates were designated as controls (referred to as *Cre*^SMC). A total of forty-seven *Il11*^SMC mice (tam-treated $n = 35$; veh-treated $n = 12$) and twenty-seven *Cre*^SMC mice were used. Individual mice died due to bowel inflammation and bleeding ($n = 7$) or were humanely euthanized when mice showed signs of pronounced weight loss and rectal prolapse ($n = 15$).

For genotyping of mice genomic DNA, we performed polymerase chain reaction (PCR) on the tail biopsies which were obtained at the time of weaning. Genotyping was conducted in two sequential PCRs, for *Myh11-Cre* and *Rosa26-Il11* genes separately. Agarose gel electrophoresis was subsequently conducted to confirm the respective product sizes for genotyping. Genotyping primer sequences are listed in S1 Table.

**Fibroblast-specific Il11 transgenic model.**  To model fibroblasts-selective secretion of IL11 *in vivo*, we crossed the heterozygous *Rosa26-IL11* mice with *Col1a2-CreER* mice [16] to generate double heterozygous *Col1a2-CreER:Rosa26-Il11* mice (referred to as *Il11*^Fib) [9]. For Cre-mediated *Il11* transgene induction, *Il11*^Fib mice were intraperitoneal injected with 50 mg kg$^{-1}$ tamoxifen at 6 weeks of age for 10 consecutive days and the animals were sacrificed on day 21. Wildtype littermates (designated as control) were injected with an equivalent dose of tamoxifen for 10 consecutive days as controls. Both female and male mice were used.

Colon length was measured from the caecum to the anus. The most distal half was taken for histology and the adjacent part was portioned and immediately snap frozen in liquid nitrogen for downstream molecular work (hydroxyproline assay, western blot analysis and quantitative polymerase chain reaction assessment). The excised heart was halved from the base to mid ventricle for histology and the remainder separated into 3 portions for molecular work. The left lung was isolated for histology and the right lung separated into 3 portions for molecular work. The right lobe of the liver was excised for histology and the left lobe separated into 3 portions for molecular work. The left kidney was fixed for histology and the right kidney separated in thirds for molecular work. The dorsal skin was harvested and halved for histology and molecular work.

## Hydroxyproline assay

The amount of total tissue collagen was quantified using colorimetric detection of hydroxyproline using the Quickzyme Total Collagen assay kit (Quickzyme Biosciences) performed according to the manufacturer's protocol. All samples were run in duplicate and absorbance at 570 nm was detected on a SpectraMax M3 fluorescence microplate reader using SoftMax Pro version 6.2.1 software (Molecular Devices).

## Fecal calprotectin (S100A8/A9) levels

To characterize inflammation in the gut, we investigated levels of fecal calprotectin in the *Il11*^SMC and *Il11*^Fib mice using the mouse S100A8/A9 heterodimer duoset ELISA kit (DY8596-05, R&D systems). Calprotectin is a biomarker for inflammatory activity and has been clinically applied as a diagnostic tool for inflammatory bowel diseases [17, 18]. Stool samples were collected in a 1.5 ml tube and diluted with 50x (weight per volume) of extraction buffer (0.1 M

Tris, 0.15 M NaCl, 1.0 M urea, 10 mM CaCl2, 0.1 M citric acid monohydrate, 5 g/l BSA (pH 8.0)) with the assumption of fecal density to be 1 g/ml. Samples were homogenized until no large particles were present. Homogenate was transferred into a fresh tube and further centrifuged at 10,000 g at 4 °C for 20 minutes. The supernatant was assessed for S100A8/A9 levels by ELISA as per the manufacturer's instructions.

## RT-qPCR

Total RNA was extracted from snap-frozen tissues using RNAzol RT (R4533, Sigma-Aldrich) followed by Purelink RNA mini kit (12183025, Invitrogen) purification. The cDNA was prepared using iScript cDNA synthesis kit (1708891, Bio-Rad) as per the manufacturer's instructions. Quantitative RT-PCR gene expression analysis was performed on duplicate samples using fast SYBR green (Qiagen) technology using the ViiA 7 Real-Time PCR System (Applied Biosystem). RT-qPCR primers are listed in S2 Table. Expression data were normalized to *Gapdh* mRNA expression levels and the $2^{-\Delta\Delta CT}$ method was used to calculate the fold change.

## Immunoblotting

Western blots were carried out on total protein extracts from mouse tissues. Frozen tissues were homogenized and lyzed in radioimmunoprecipitation assay (RIPA) buffer containing protease and phosphatase inhibitors (Roche) followed by centrifugation. Equal amounts of protein lysates were separated by SDS-PAGE, transferred onto PVDF membrane and immunoblotted for pERK1/2 (4370, CST), ERK1/2 (4695, CST), pSTAT3 (4113, CST), STAT3 (4904, CST), GAPDH (2118, CST) and IL11 (X203, Aldevron). Proteins were visualized with appropriate secondary antibodies anti-rabbit HRP (7074, CST) and anti-mouse HRP (7076, CST).

## Histology

Tissues from *Il11*<sup>SMC</sup> and *Il11*<sup>Fib</sup> mice were fixed in 10% neutral-buffered formalin for 24–48 hours, tissue processed and paraffin-embedded. Sections were obtained at 5 μm and stained with Masson's trichrome staining for collagen. Brightfield photomicrographs of the sections were randomly captured by a researcher blinded to the treatment groups using the Olympus BX51 microscope and Image-Pro Premier 9.2 (Media Cybernetics).

Photomicrographs of the colon taken at 200X magnification were used to calculate muscle wall thickness. The distance between the inner and outer circumference of the muscularis propria was measured using the incremental distance tool at a calibrated step size of 25 μm on Image-Pro Premier 9.2 (Media Cybernetics). A total of 75 to 250 measurements across three to five photomicrographs per section were taken and averages reported per photomicrograph. Muscle thickness was reported as an average across 3 cross-sections of the colon per animal.

Photomicrographs of the dorsal skin were captured in 3 fields per section at 100X magnification and used to calculate epidermal and dermal thickness. The epidermis was measured from the stratum basale to the stratum granulosum using hand-drawn line segments on Image-Pro Premier 9.2 (Media Cybernetics). The dermis was measured from the dermal-epidermal junction to the hypodermis. Measurements were recorded using the incremental distance tool at a calibrated step size of 50 μm on Image-Pro Premier 9.2 (Media Cybernetics). A total of 75 to 200 measurements across three photomicrographs per section were taken and averages reported per photomicrograph. Overall epidermal and dermal thickness was reported as an average across the 3 fields per animal.

Fibrosis quantification was conducted as referenced [19]. Color deconvolution version 1.5 plugin using the Masson Trichrome vector on ImageJ (version 1.52a, NIH) and thresholding was applied for area quantification. Perivascular fibrosis was measured as a ratio of the fibrosis

area to the vessel area. Vascular hypertrophy was quantified as the ratio of media wall area to the lumen area.

## Immunohistochemistry

Paraffin-embedded colon tissue were sectioned at 5 μm, deparaffinized and permeabilized with Triton X-100 (Sigma-Aldrich) and heat antigen retrieved with Bull's Eye Decloaker (Biocare Medical). Slides were then blocked for endogenous peroxidase with Bloxall™ blocking solution (Vector Lab) followed by blocking with either 3% bovine serum albumin, or mouse on mouse blocking reagent (Vector Lab). Anti-IL11 (ab10558; PA5-36544, Invitrogen), anti-CD45 (1:100; ab10558, Abcam), anti-LGALS3 (1 μg/ml; CL8942AP, Cedarlane) amd anti-LAMP2 (1 μg/ml; 550292, BD Bioscience) were added and incubated overnight at 4°C. Anti-rabbit (1:100; ab27478, Abcam) and anti-rat IgG (1 μg/ml; sc-2026, Santa Cruz) isotype controls were added as respective negative controls. Slides were incubated with anti-rabbit IgG peroxidase (1:500, A0545, Sigma-Aldrich) and anti-rat IgG peroxidase (MP-7404, Vector Lab) followed by chromogen development with ImmPACT® DAB peroxidase substrate kit (SK-4105, Vector Lab) according to manufacturer's instructions. Lastly, Gill's haematoxylin (H-3401, Vector Lab) was added for nuclear counterstain. To control for unspecific binding, primary antibody isotype controls were included and images are presented in S3 Fig.

## Statistical analysis

Data are presented as mean ± standard deviation or median ± range as stated in the figure legends. Statistical analyses were performed on GraphPad Prism 8 software (version 8.1.2). Outliers (ROUT 2%, GraphPad Prism software) were removed prior to analyses. Comparison of survival curves was analyzed with the log-rank Mantel-Cox test. Bodyweight progression was analyzed with two-way ANOVA with Sidak multiple comparisons. A comparison of mice strains for all other parameters was analyzed with a two-tailed unpaired $t$-test. The criterion for statistical significance was established at $P < 0.05$.

## Results

### Expression of Il11 in smooth muscle cells results in ill health and early mortality

We generated an $Il11^{SMC}$ mouse model that overexpresses IL11 specifically in $Myh11^{+ve}$ SMCs: Conditional transgenic mice with mouse $Il11$ inserted into the Rosa26 locus ($Rosa26$-$Il11$-Tg) [8] were crossed with smooth muscle-specific $Myh11$-$cre/ERT2$ mice [15] (Fig 1a and 1b). We then injected tamoxifen (tam) three times at day 0, 3 and 5 into 6-week old $Il11^{SMC}$ mice to induce recombination in $Myh11^{+ve}$ cells and monitored the survival and body weight for 14 days. Following tam-induced $Il11$ expression in SMCs, mice started dying from day three onwards, with only 37% of $Il11^{SMC}$ mice surviving to day 14. This was significantly different from the survival of either vehicle (veh)-treated $Il11^{SMC}$ animals or tam-treated $Cre^{SMC}$ control mice, which were unaffected and both had 100% survival (both $P < 0.001$; Fig 1d and S1b Fig). Starting from day four onwards, tam-treated $Il11^{SMC}$ mice progressively lost weight as compared to veh -treated and tam-treated $Cre^{SMC}$ controls (both $P < 0.001$; Fig 1e and S1d Fig). Following two weeks of tam-induced $Il11$ expression, $Il11^{SMC}$ mice were significantly smaller in body weight and length as compared to tam-treated $Cre^{SMC}$ controls (both $P < 0.001$; S1f and S1g Fig) and veh-treated $Il11^{SMC}$ mice ($P = 0.002$ and $P < 0.001$ respectively; Fig 1f and 1g). In contrast, the indexed weight of the heart, lung and kidney in tam-treated $Il11^{SMC}$ animals was significantly elevated ($P_{Heart} < 0.001$; $P_{Lung} < 0.001$; $P_{Kidney} =$

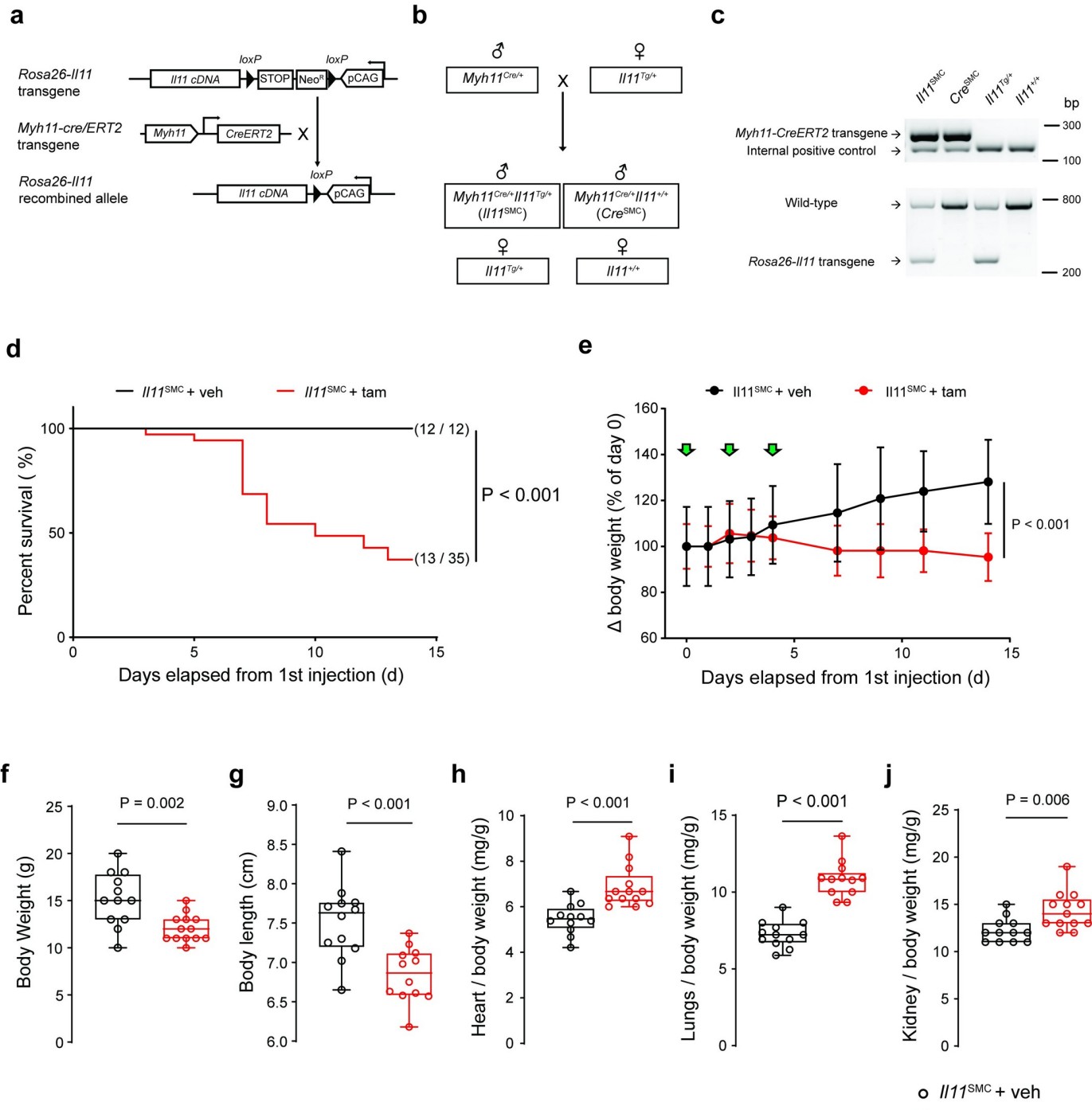

**Fig 1. Expression of *Il11* in smooth muscle cells is associated with body weight loss, elevated organ weights and spontaneous death. (a)** Schematic diagram of the targeted expression of *Il11* in *Myh11*[+ve] SMC. In *Rosa26-Il11* mice, a floxed cassette containing both the neomycin (neo) resistance and stop elements is positioned before the murine *Il11* transgene cassette, which undergoes tamoxifen (tam) initiated *Cre*-mediated recombination when crossed to the *Myh11-Cre/ERT2* mouse. **(b)** Breeding scheme to generate *Myh11*[Cre/+]*Rosa26*[Il11/+] (*Il11*[SMC]) and *Myh11*[Cre/+]*Rosa26*[+/+] (*Cre*[SMC]) offspring mice. Note that the *Myh11-Cre* gene is expressed on the Y chromosome and therefore only male offspring carry the transgene. **(c)** Genotyping of tail biopsy DNA. A 287 bp band indicates the presence of the *Cre* transgene whereas the 180 bp band determines the presence of the internal positive control (top gel). Polymerase chain reaction with the *Rosa26-Il11* primer set detects a 270 bp band indicative of the *Rosa26-Il11* transgene whereas the 727 bp band indicates the presence of the wild-type transgene (bottom gel). Uncropped blots are presented in S2 Fig. **(d)** Survival curve of *Il11*[SMC] mice treated with tam (*n* = 35) and corn oil vehicle (veh; *n* = 12) mice following tamoxifen initiation at day 0 and followed until day 14. Survival curves were compared using the log-rank Mantel-Cox test. **(e)** Body weight changes (expressed as percentage of day 0 body weight) in *Il11*[SMC] mice treated with tam or veh (*n* = 8 per group). Green arrows denote individual injections. Statistical analyses by two-way

ANOVA with Sidak multiple comparisons; data expressed as mean ± standard deviation. **(f)** Collated body weights (left) and **(g)** body lengths (right) of *Il11*^SMC^ mice treated with tam or veh measured at d14 post initial tamoxifen dose (*n* = 12–13 per group). **(h)** Organ weights of the heart, **(i)** lung and **(j)** kidney normalized to body weight in *Il11*^SMC^ mice treated with tam or veh (*n* = 12–13 per group). All comparisons were conducted in mice 14 days post-veh and tam treatment. Statistical analyses by two-tailed unpaired t-test; data expressed as median ± IQR, whiskers represent the minimum and maximum values.

0.006) when compared to veh-treated mice (Fig 1h). We did not observe differences in liver weight or colon length in veh or tam treated *Il11*^SMC^ animals (data not shown).

## IL11 expression causes severe inflammatory bowel disease associated with fibrosis

The most obvious and striking feature of *Il11*^SMC^ mice treated with tam was progressive rectal prolapse and pale loose stool formation from as early as day three after gene induction (Fig 2a and S1c Fig). Gross anatomical inspection of the gastrointestinal tract revealed inflammation and swelling of the small and large intestines of tam-treated *Il11*^SMC^ mice when compared to veh-treated controls (Fig 2b). Intestinal inflammation was specifically indicated by an increase in fecal calprotectin, a biomarker used to monitor disease activity in human colitis, in tam-treated *Il11*^SMC^ mice when compared to veh treatment (P < 0.001; Fig 2c). Masson's trichrome staining of the colon indicated a very large increase in collagen deposition (P < 0.001; Fig 2d and 2e). Histology also showed a significant increase in the thickness of the smooth muscle-dominant muscularis propria (P = 0.040; Fig 2f). Quantitative hydroxyproline assessments revealed an increase in colonic collagen content in *Il11*^SMC^ mice after tam treatment (P < 0.001; Fig 2g), confirming the histological data.

We then performed an immunohistochemical staining for IL11, CD45, lysosomal-associated membrane protein 2 (LAMP2) and lectin, galactose binding, soluble 3 (LGALS3) in the smooth muscle and crypt compartment of the colon in veh and tam-treated *Il11*^SMC^ mice (Fig 2h). In *Il11*^SMC^ mice, IL11 staining was diffuse in the smooth muscle, perhaps with a higher background staining, and also localized more strongly to other stromal cells that are likely fibroblasts, which express the IL11 receptor [8]. Furthermore, CD45^+ve^ leukocytes were increased in the fibrotic regions and crypts of the tam-treated *Il11*^SMC^ mouse colon. LAMP2 and LGALS3 are markers for epithelial cells and activated macrophages contributing to intestinal inflammation. Tam-treated *Il11*^SMC^ colon demonstrated increased expression of LAMP2 and LGALS3 in the epithelial cells and leukocytes of the crypts, consistent with inflammation in these regions [20–23]. In tam-treated *Il11*^SMC^ treated with Tam as compared to veh-treated mice, there was also activation of leukocytes in the Peyer's patches of the colon, which are a primary site of mucosal immune response (Fig 3a), as well as in localized areas of disrupted villi architecture (Fig 3b). Interestingly, the myenteric plexus of the colon demonstrated ganglionic hyperplasia and fibrosis (Fig 3c), typical of neuroinflammation associated with inflammatory bowel disease.

## IL11 expression in smooth muscle cells activates non-canonical IL11 signaling pathways

Given that smooth muscle cells are expressed in the walls of most organs, including the vasculature, bronchi, gastrointestinal and abdominal organs, we sought to confirm the expression of *Il11* in *Il11*^SMC^ mice across tissues and performed western blotting at 14 days after tamoxifen administration. This confirmed that IL11 protein was significantly upregulated at the protein level across all tissues tested ($P_{colon}$ = 0.034; $P_{heart}$ = 0.002; $P_{lung}$ = 0.039; $P_{liver}$ < 0.001; $P_{kidney}$ = 0.004; and $P_{skin}$ = 0.004; Fig 4).

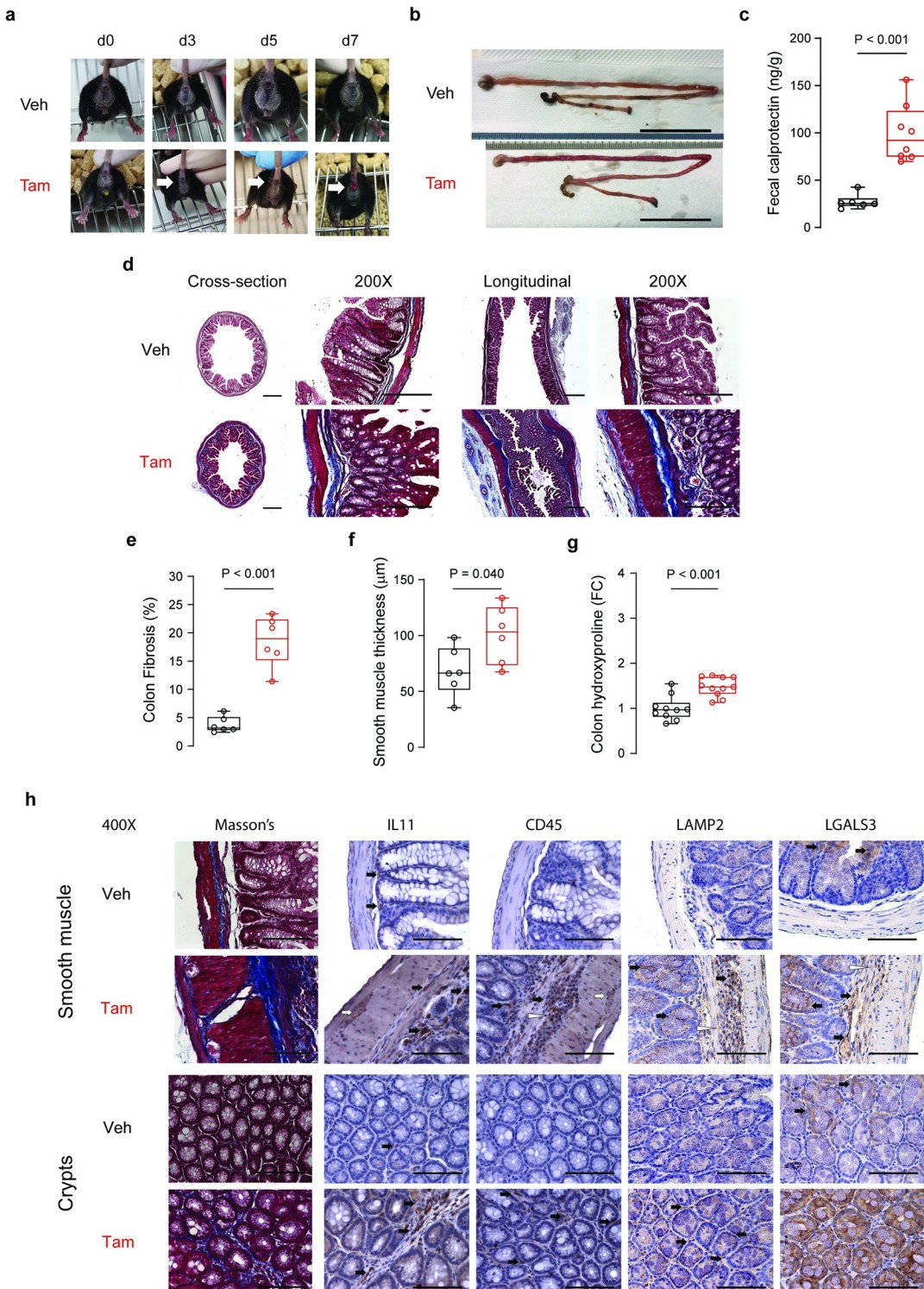

**Fig 2. *Il11* expression results in fibro-inflammatory disease of the colon. (a)** Representative images of the *Il11*^SMC mice before (d0) and up to 7 days (d7) treatment with either corn oil vehicle (veh) or tamoxifen (tam). Presence of rectal prolapse are indicated with white arrows. Images represent the same animal across time points not taken to the same scale. **(b)** Excised gastrointestinal tract of representative *Il11*^SMC mice at day 14 post-treatment with veh or tam. Scale bar represents 5 cm. **(c)** Fecal calprotectin in representative *Il11*^SMC mice treated with veh or tam assessed by ELISA (*n* = 6–8 per group). **(d)** Representative cross-section and longitudinal section of the colon stained with Masson's trichrome (left) and at 200X magnification (right). Scale bar of cross and longitudinal sections represents 500 μm and at 200X magnification represents 200 μm. **(e)** Colon fibrosis determined as a percentage of collagen positive area (blue) from histological images taken at 200X

magnification (*n* = 6 per group). (**f**) Tunica muscularis (smooth muscle) thickness of the colon (*n* = 6 per group). (**g**) Total collagen content assessed by hydroxyproline assay and expressed as fold change (FC) of veh-treated *Il11*$^{SMC}$ mice (*n* = 10–11 per group). (**h**) Representative images of the colonic smooth muscle and crypts taken at 400X magnification for Masson's trichrome and immunohistochemistry staining for IL11, cluster of differentiation 45 (CD45), lysosome-associated membrane protein 2 (LAMP2), and galectin-3 (LGALS3) (*n* = 3 per group). Black arrows denote focal staining of positive cells, white arrows denote myenteric plexus which are positive for IL11 and CD45 expression, and white arrowheads denotes leukocyte aggregation. Scale bars represent 100 μm. All comparisons were conducted in organs harvested from mice 14 days post-veh and tam treatment. Statistical analyses by two-tailed unpaired t-test; data expressed as median ± IQR, whiskers represent the minimum and maximum values.

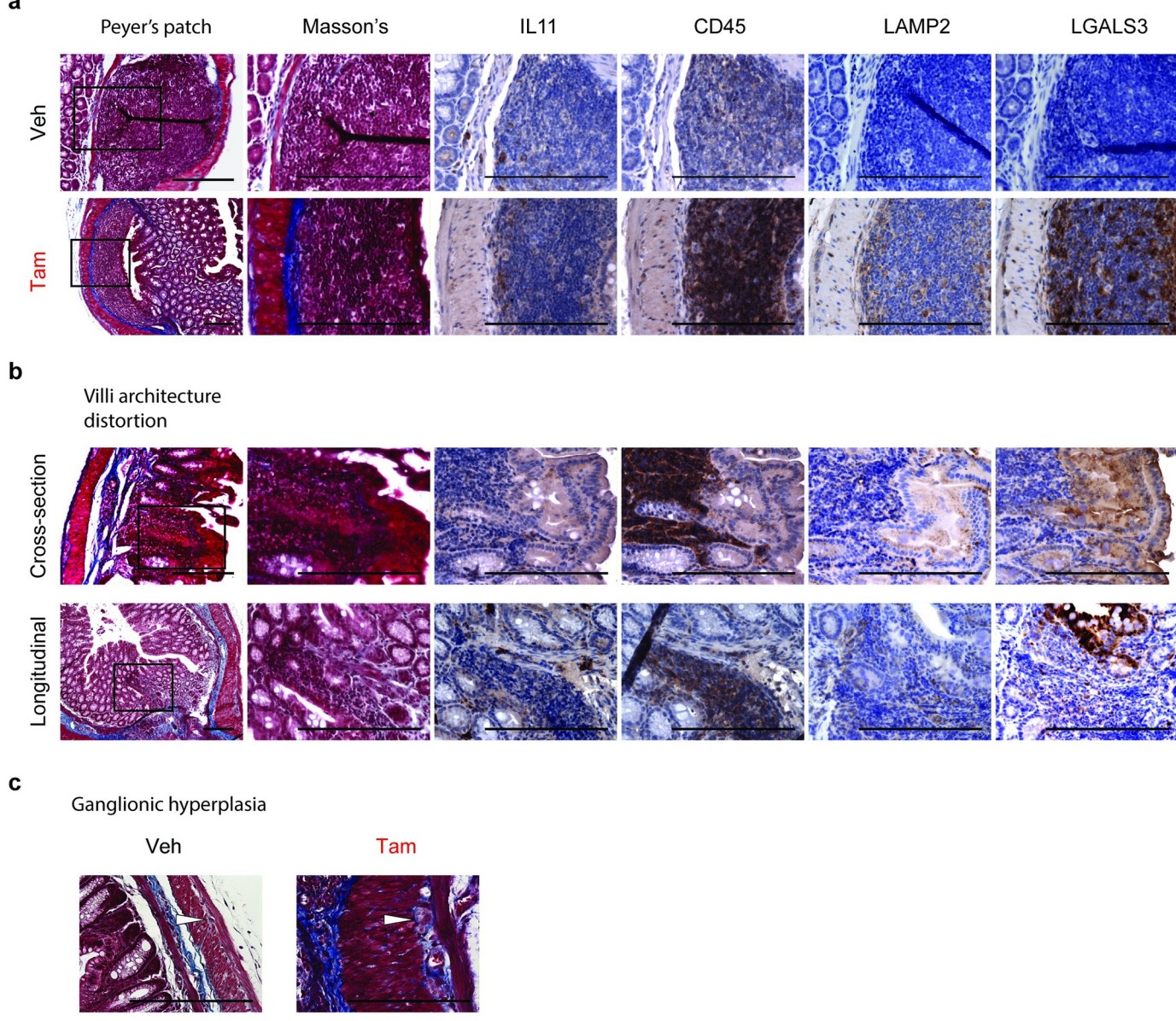

**Fig 3. *Il11* expression in smooth muscle cells leads to lymphoid cell aggregates, villi distortion and ganglionic hyperplasia.** (**a**) Peyer's patch of tam-treated *Il11*$^{SMC}$ mice showed increased expression of IL11, CD45, LAMP2 and LGALS3 compared to veh-treated controls. (**b**) Cross- and longitudinal sections of the mucosa region of the colon in tam-treated *Il11*$^{SMC}$ mice have inflammatory cell infiltrates that extend from the submucosa to the mucosa region resulting in distortion of villi architecture. (**c**) Ganglionic hyperplasia and fibrosis in the myenteric plexus of tam-treated *Il11*$^{SMC}$ mice compared to vehicle controls. White arrowheads denote ganglionic cells. Black boxes were imaged at 400X magnification. All scale bars represent 200 μm.

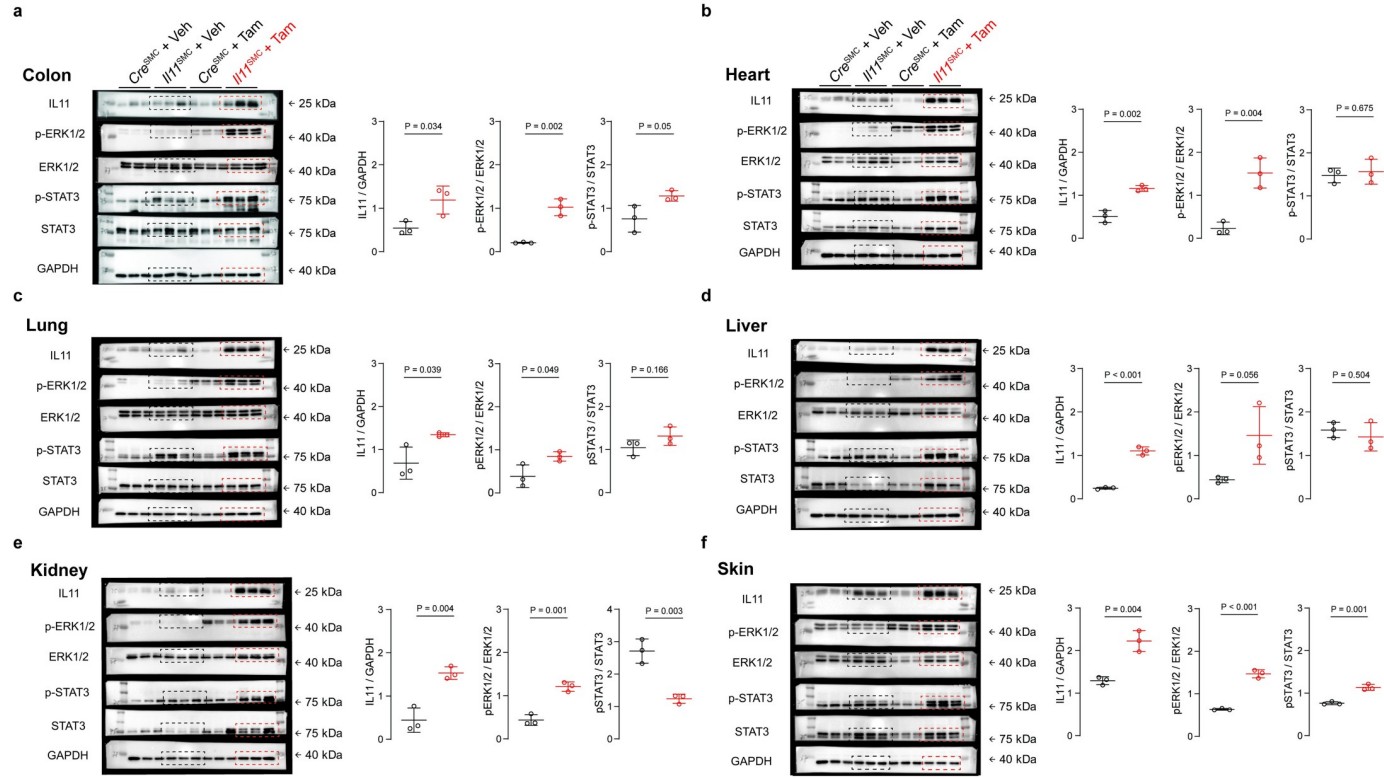

**Fig 4. *Il11*<sup>SMC</sup> mice exhibit activated ERK1/2 signaling across organs.** Immunoblots of IL11 expression, phospho- (p) and total ERK1/2 and STAT3 protein in **(a)** colon, **(b)** heart, **(c)** liver, **(d)** lung, **(e)** kidney, **(f)** skin tissue of *Il11*<sup>SMC</sup> mice treated with vehicle (veh) or tamoxifen (tam) (*n* = 3 per group). Dotted boxes in the immunoblots represent the veh-treated (black) and tam-treated (red) groups for all organs. All comparisons were conducted in organs harvested from mice 14 days post-veh or tam treatment.

IL11 is a member of the IL6 family of cytokines, which are considered to signal via the Janus Kinase (JAK)/Signal Transducer and Activator of Transcription (STAT) pathway [24]. However, we recently showed that the IL11 effect, both *in vitro* in fibroblasts and *in vivo* at the tissue level, is also dependent on non-canonical signaling via extracellular signal-regulated kinase (ERK) [8–10]. To investigate both canonical and non-canonical signaling pathways after *Il11* expression, we performed western blotting of phosphorylated (p) STAT3 or ERK1/2 and total protein levels and derived indices of kinase activation by normalizing phosphorylation amounts to total protein levels (Fig 4). At baseline, ERK was phosphorylated at low levels in most tissues except for the skin. Upon IL11 expression, we detected a strong and significant activation of ERK in all tissues ($P_{colon}$ = 0.002; $P_{heart}$ = 0.004; $P_{lung}$ = 0.049; $P_{liver}$ = 0.056; $P_{kidney}$ = 0.001; and $P_{skin}$ < 0.001; Fig 4). STAT3 phosphorylation was unchanged in the heart, lung and liver but was elevated in the colon and skin (P = 0.05 and 0.001 respectively; Fig 4). In contrast, total levels of STAT3 appeared to be increased in the liver and kidney of tam-treated *Il11*<sup>SMC</sup> animals (Fig 4d and 4e). Overall, while both pathways were affected, ERK signaling was consistently activated across tissues tested whereas STAT3 was not.

## IL11 destroys tissue integrity and promotes collagen deposition

To investigate the effect of *Il11* expression in SMCs on tissue composition beyond the colon, we performed histological analyses of the heart, lung, liver, kidney and skin. Masson's trichrome staining was used to visualize collagen and quantify extracellular matrix deposition. In

the heart, we observed collagen deposition in the perivascular region (P = 0.002; Fig 5a and 5b). We also observed vascular hypertrophy (P = 0.019; Fig 5c) and mild ventricular hypertrophy in the absence of dilatation (data not shown). Hydroxyproline assay of the whole heart confirmed cardiac fibrosis (P = 0.026; Fig 5d). In the lung, Ashcroft scores of pulmonary histological images showed lung damage after tam-induced *Il11* expression (P < 0.001; Fig 5e and 5f). Masson's trichrome staining indicated elevated collagen expression throughout the lung in *Il11*[SMC] mice and pulmonary fibrosis was confirmed by the hydroxyproline assay (P = 0.001; Fig 5g).

The effect of *Il11* expression on the liver was overall mild and characterized by perisinusoidal fibrosis (Fig 5h to 5j). Renal tissue structure was also affected only mildly, with limited fibrosis occurring around the blood vessels (Fig 5k to 5m). The effect of IL11 on the skin of tam-treated *Il11*[SMC] animals was more profound and both the dermal and epidermal thickness was significantly increased (Fig 5n to 5p; P = 0.041 and P = 0.001 respectively). Dorsal skin sections showed that epidermal and dermal cell infiltrates were increased and the adipose tissue layer in the hypodermis was largely depleted. Confirming Masson's trichrome staining of skin sections, we observed increased collagen deposition in the skin of tam-treated *Il11*[SMC] mice using the hydroxyproline assay (P < 0.001; Fig 5q).

## IL11 secretion from smooth muscle cells drives fibrogenic gene expression

We assessed the RNA expression of fibrogenic genes to complement our histology studies, which further substantiated the presence of multi-organ fibrosis. Reverse transcription-polymerase chain reaction (RT-PCR) was performed using RNA from colonic, ventricular, pulmonary, hepatic, renal and skin tissue of veh- or tam-treated *Il11*[SMC] mice. Collagen, type I, alpha 1 (*Col1a1*) RNA was significantly upregulated in all tissues ($P_{colon}$ = 0.005; $P_{heart}$ = 0.005; $P_{lung}$ < 0.001; $P_{liver}$ = 0.016; $P_{kidney}$ = 0.022; $P_{skin}$ = 0.016; Fig 6), confirming the effect of *Il11* expression on global organ fibrosis that we observed on the protein level (Fig 5). Additional markers for fibrosis such as collagen, type I, alpha 2 (*Cola1a2*), collagen, type III, alpha 1 (*Col3a1*), fibronectin 1 (*Fn1*), tissue inhibitor of metalloproteinase 1 (*Timp1*) and matrix metallopeptidase 2 (*Mmp2*) were also assessed via RT-PCR (Fig 6a to 6f). These genes were elevated in most tissues of tam-treated *Il11*[SMC] mice. *Timp1* transcripts were significantly upregulated in the heart (P < 0.001), lung (P = 0.004), liver (P = 0.003), kidney (P = 0.003) and skin (P = 0.017), which is a recognized feature of pathological ECM remodeling [25].

## IL11 secreted from smooth muscle cells causes inflammation across tissues

In addition to fibrosis, SMC-driven diseases are often characterized by tissue inflammation. To better understand whether IL11 secretion from SMCs can contribute to this pathology, we performed RT-PCR experiments of inflammatory marker genes across multiple tissues. Interleukin 6 (IL6) also signals via gp130, similar to IL11, but its specific IL6 receptor subunit is expressed on a different subset of cells, most of which belong to the immune system [8]. IL6 is also a well-established therapeutic target for inflammatory diseases such as rheumatoid arthritis [26]. Upon tam-induced *Il11* expression in *Il11*[SMC] mice, we found *Il6* mRNA to be significantly upregulated across all tissues tested ($P_{colon}$ = 0.001; $P_{heart}$ < 0.001; $P_{lung}$ = 0.015; $P_{liver}$ = 0.007; $P_{kidney}$ < 0.001; and $P_{skin}$ = 0.003; Fig 7).

In the colon, we also detected increased RNA expression of the inflammatory chemokine C-C motif chemokine ligand 2 (*Ccl2*) (P = 0.017), whereas C-C motif chemokine ligand 5 (*Ccl5*) was not significantly elevated but trended upwards (P = 0.141). Interestingly, these inflammatory chemokines are upregulated in the colonic mucosa of IBD patients [27, 28]. However, *CCL2* transcripts, and not *CCL5* transcripts, were found to be expressed in vessel-

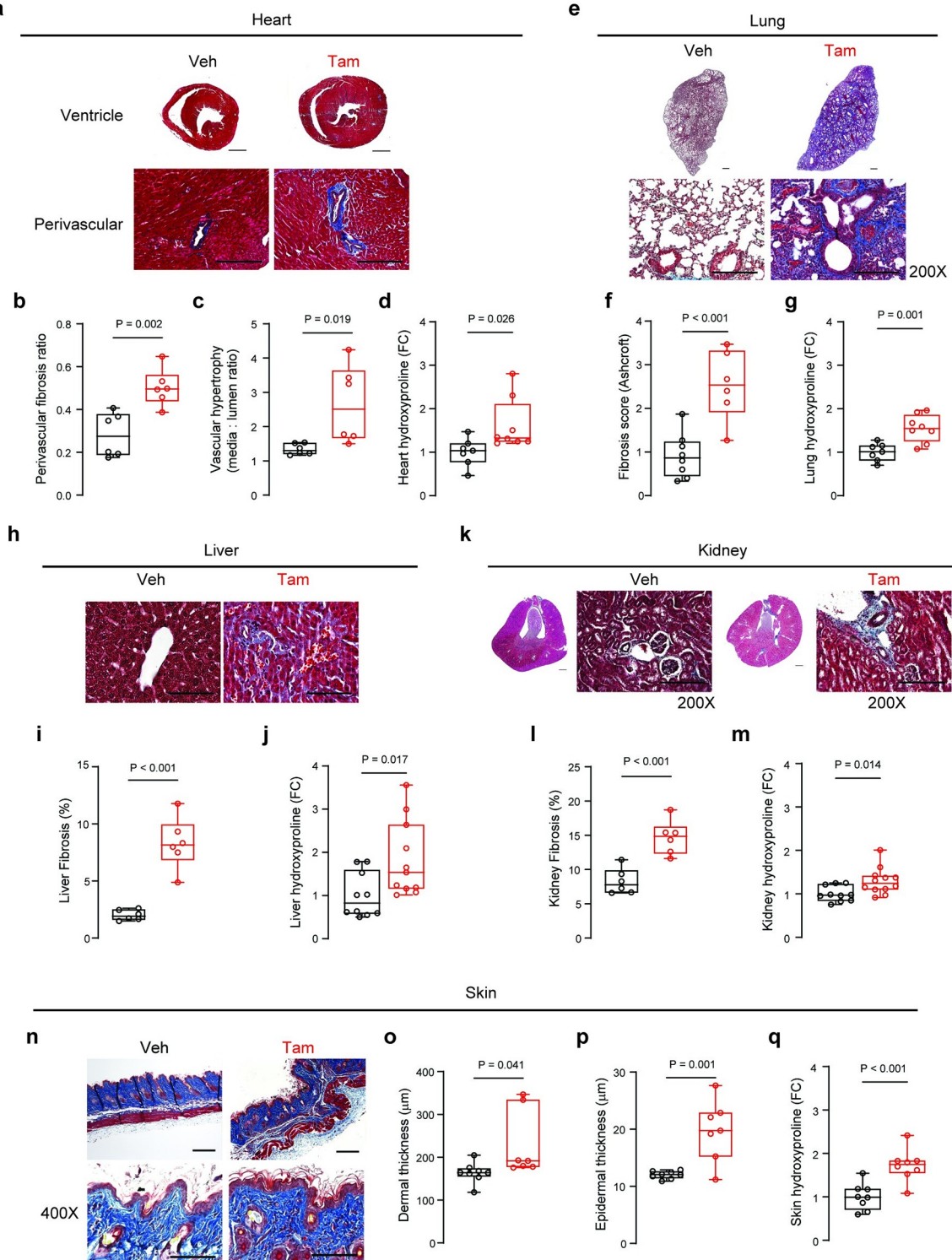

**Fig 5. *Il11* expression in smooth muscle cells causes fibrosis across organs. (a)** Representative Masson's trichrome stained mid-ventricle sections of the heart harvested at 14 days post-vehicle (veh) or tamoxifen (tam) initiation (left) and 200X magnification images demonstrating perivascular fibrosis (right). Scale bars for mid-ventricle sections and 200X magnification denote 500 μm and 200 μm respectively. **(b)** Perivascular fibrosis quantification of histological images from veh- and tam-treated *Il11*$^{SMC}$ mice at 200X magnification (*n* = 6 per group). **(c)** Vascular hypertrophy quantification of veh- and tam-treated *Il11*$^{SMC}$ mice (*n* = 6 per group). **(d)** Total collagen content in the heart assessed by hydroxyproline assay and shown as fold change (FC) of veh- and tam-treated *Il11*$^{SMC}$

mice (*n* = 7–8 per group). **(e)** Representative Masson's trichrome stained whole lung sections (left) and 200X magnification images (right). Scale bars for whole lung sections and 200X magnification denote 500 μm and 200 μm respectively. **(f)** Pulmonary fibrosis quantification as assessed by the Ashcroft score (*n* = 6–8 per group). **(g)** Total collagen content in the lung assessed by hydroxyproline assay as above (*n* = 7–8 per group). **(h)** Representative Masson's trichrome stained liver sections taken at 400X magnification demonstrating perisinusoidal fibrosis. Scale bar at 400X magnification indicates 100 μm. **(i)** Fibrosis quantification of liver sections (400X magnification) from veh- and tam-treated *Il11*^SMC mice (*n* = 6 per group). **(j)** Total collagen content in the liver assessed by hydroxyproline assay as above (*n* = 10–11 per group). **(k)** Representative Masson's trichrome stained cross-section of the kidney (left) and 200X magnification images (right). Scale bars for the cross-section of the kidney and 200X magnification denote 500 μm and 200 μm respectively. **(l)** Fibrosis quantification of kidney sections (200X magnification) from veh- and tam-treated *Il11*^SMC mice (*n* = 6 per group). **(m)** Total collagen content in the kidney assessed by hydroxyproline assay as above (*n* = 10–11 per group). **(n)** Representative Masson's trichrome stained section of the dorsal skin at 100X magnification (left) and at 400X magnification (right). Scale bar at 100X and 400X magnification represents 200 μm and 100 μm respectively. **(o)** Dermal and **(p)** epidermal thickness of the dorsal skin. **(q)** Total collagen content in the skin assessed by hydroxyproline assay as above (*n* = 10–11 per group). All comparisons were conducted in organs harvested from mice 14 days post-veh and tam treatment. Statistical analyses by two-tailed unpaired t-test; data shown as median ± IQR, whiskers represent the minimum and maximum values.

associated cells such as SMCs in IBD [27]. Given that *Il11*^SMC mice express *Il11* in SMCs, it is consistent that the transcript expression of the chemokine expressed in this particular cellular niche in the colon is most affected. In the skin, all three inflammatory markers tested were highly upregulated. This points to an inflammatory gene expression signature in the skin that is reminiscent of that seen in systemic sclerosis, since *IL6*, *CCL2*, and *CCL5* are elevated in the serum of patients [29, 30]. Of note, *CCL2* levels were correlated with the extent of skin fibrosis in systemic sclerosis, a pathogenic feature also triggered by IL11 expression in SMCs (Fig 7) [29].

### Fibroblast-selective expression of *Il11* recapitulates the features of colonic inflammatory phenotype seen in Il11^SMC mice

We have previously described a model of *Il11* expression in fibroblasts (*Il11*^Fib) that drives fibrosis in the heart, kidney, and lung [8, 9]. To examine further the effect of *Il11* expression in stromal cells on the colon, we studied colonic phenotypes in this second model of *Il11* expression from the stromal niche (Fig 8a). Gross examination of the gastrointestinal tract of *Il11*^Fib mice revealed macroscopic appearances consistent with inflammation of the colon to a similar extent as in *Il11*^SMC mice (Fig 8b). The total gastrointestinal gut length of *Il11*^Fib mice was unchanged overall but the colon length alone was reduced (P = 0.030; Fig 8b to 8d), which is a feature of experimental colitis in mice [31]. In this model, as compared to *Il11*^SMC mice, we detected *Il6* but not *Ccl2* or *Ccl5*, upregulation in the colon (Fig 8d and 8e). Inflammation of the gut was apparent in the *Il11*^Fib model as fecal calprotectin was significantly elevated (P = 0.003; Fig 8f). Histological examination revealed marked colonic dilation and increased SMC thickness (Fig 8g and 8h). In contrast to the *Il11*^SMC model of *Il11* expression, colonic fibrosis as determined by histology, hydroxyproline assay or ECM gene expression, was not significantly different between tam-treated *Il11*^Fib and controls (data not shown). Taken together, fibroblast-driven *Il11* expression recapitulates primarily the SMC-driven inflammatory, but not the fibrotic, phenotype in the mouse.

### Discussion

In humans, *IL11* is highly upregulated in the colonic mucosa of patients with either ulcerative colitis or Crohn's disease who do not respond to anti-TNF therapy, with recent single cell RNA-seq studies localizing IL11 to inflammatory mucosal stromal cells [32–34]. To better understand the effect of IL11 in the colon, recombinant human IL11 has been used in rodent models of IBD [34–38] and it was suggested that IL11 may have a protective role in the bowel. However, a caveat with these studies is that human IL11 was administered to rodents despite

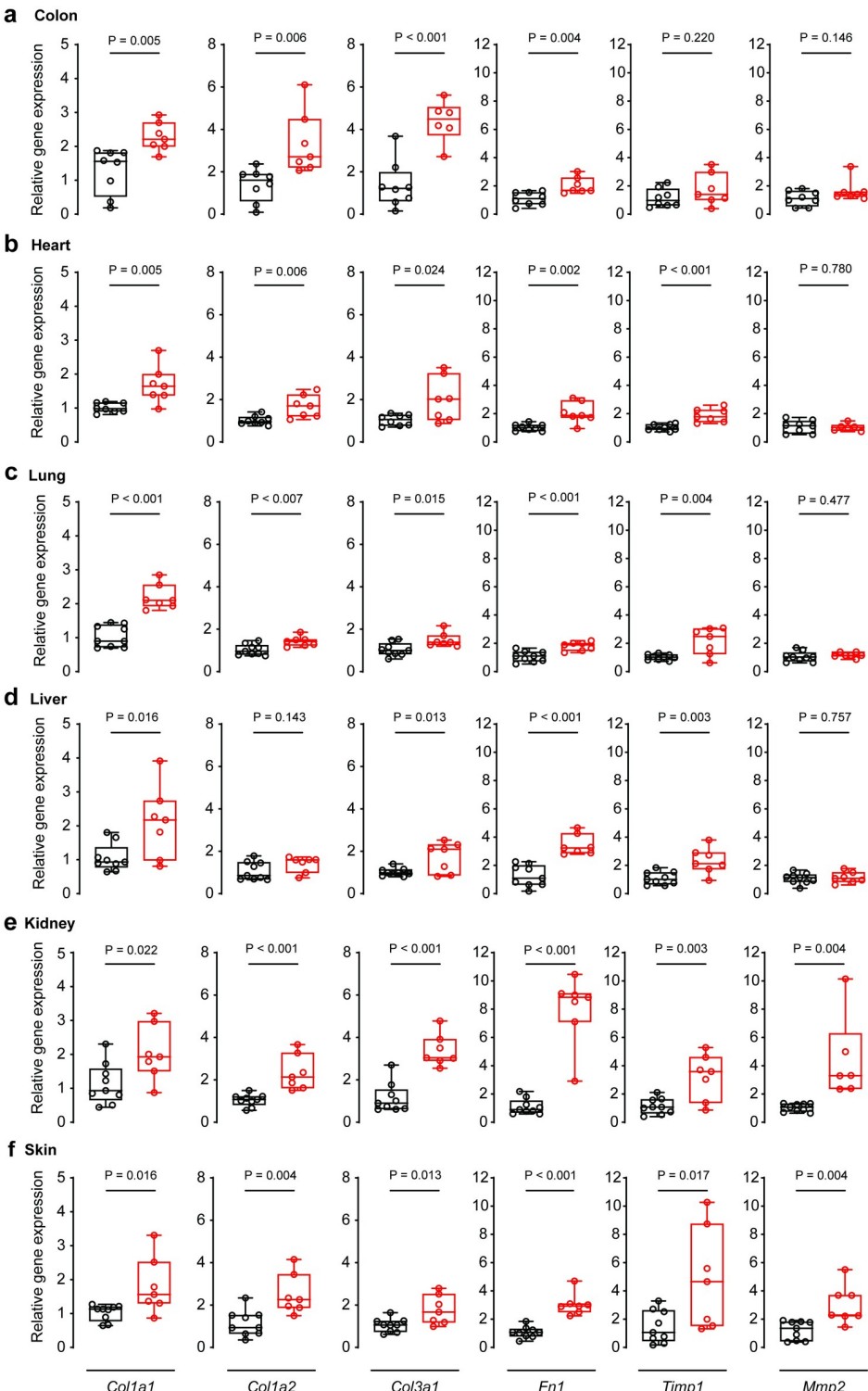

**Fig 6. Relative gene expression of fibrogenic genes in organs from tam-treated *Il11*^SMC mice.** Relative mRNA expression of collagen type 1a1 (*Col1a1*), type 1a2 (*Col1a2*), type 3a1 (*Col3a1*), fibronectin-1 (*Fn1*), tissue inhibitor of metalloproteinase 1 (*Timp1*) and matrix metalloproteinase 2 (*Mmp2*) normalized to Glyceraldehyde 3-phosphate dehydrogenase (*Gapdh*) expression in the **(a)** colon, **(b)** heart, **(c)** lung, **(d)** liver, **(e)** kidney and **(f)** skin. All comparisons were conducted 14 days post-veh (black) and tam (red) initiated mice. Statistical analyses by two-tailed unpaired *t*-test; data expressed as median ± IQR, whiskers represent the minimum and maximum values.

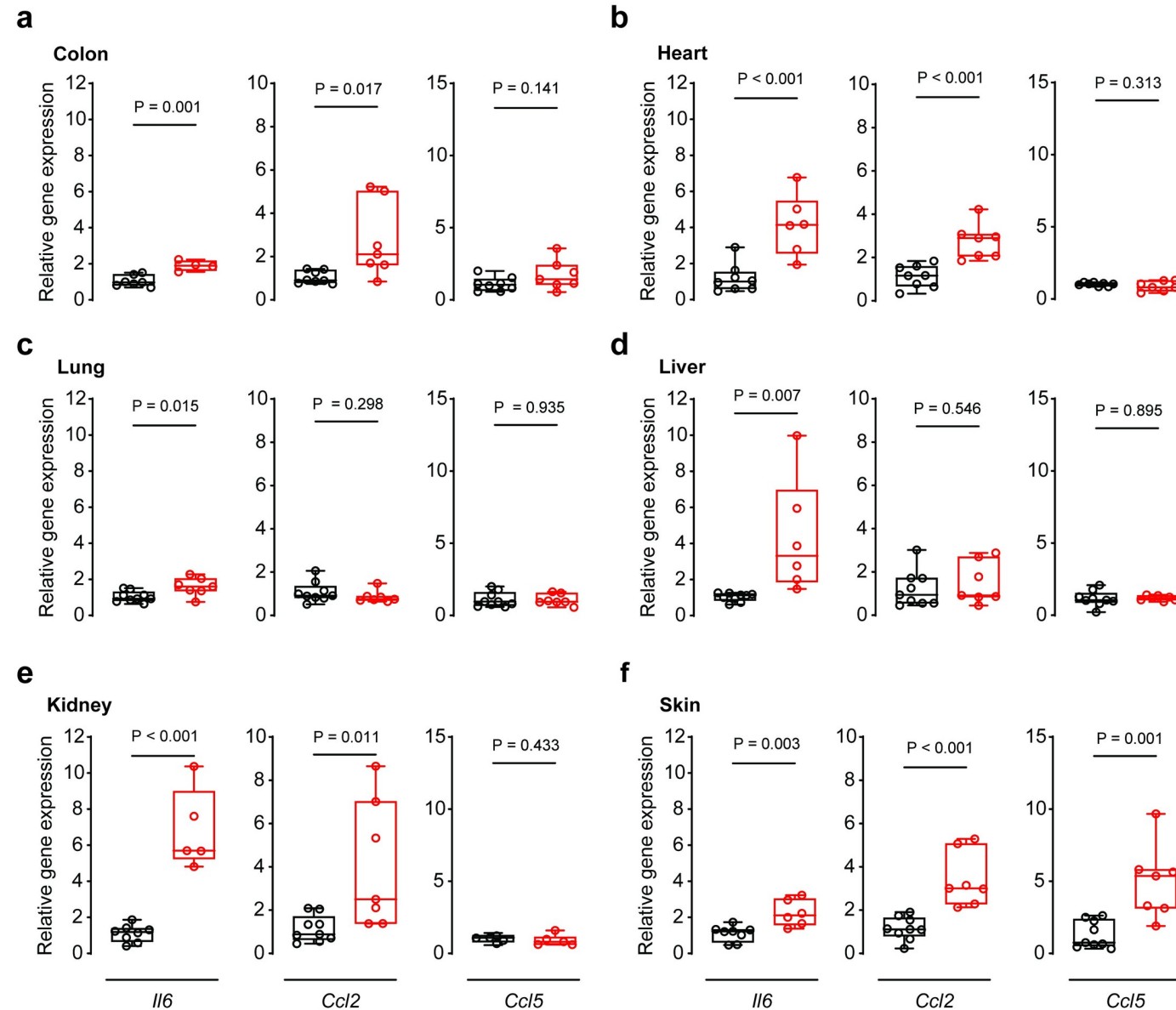

**Fig 7. Relative gene expression of inflammatory genes in organs from tam-treated *Il11*<sup>SMC</sup> mice.** Relative mRNA expression of interleukin 6 (*Il6*), C-C motif chemokine ligand 2 (*Ccl2*), C-C motif chemokine ligand 5 (*Ccl5*) normalized to Glyceraldehyde 3-phosphate dehydrogenase (*Gapdh*) expression in the **(a)** colon, **(b)** heart, **(c)** lung, **(d)** liver, **(e)** kidney and **(f)** skin respectively. All comparisons were conducted in 14 days post-veh (black) and tam (red) initiated mice. Statistical analyses by two-tailed unpaired *t*-test; data expressed as median ± IQR, whiskers represent the minimum and maximum values.

the fact that human IL11 does not activate mouse stromal cells [8]. More recently, we have found that human IL11 unexpectedly acts as an inhibitor of endogenous mouse IL11 activity in the liver [39]. Thus, previous studies in IBD that showed that when human IL11 is injected into mice it protects them from IBD may paradoxically support the opposite conclusion: IL11 is not protective at all, but a driver of IBD. In light of this, there is a great need to assess the effects of species-specific IL11 in the mouse, which we undertook in this study by expressing murine *Il11* in SMCs or fibroblasts in adult mice.

To enable our studies, we developed the *Il11*<sup>SMC</sup> mouse as a tool to study the effect of murine IL11 secreted from SMCs, an established source of IL11 in the vasculature, airway, and

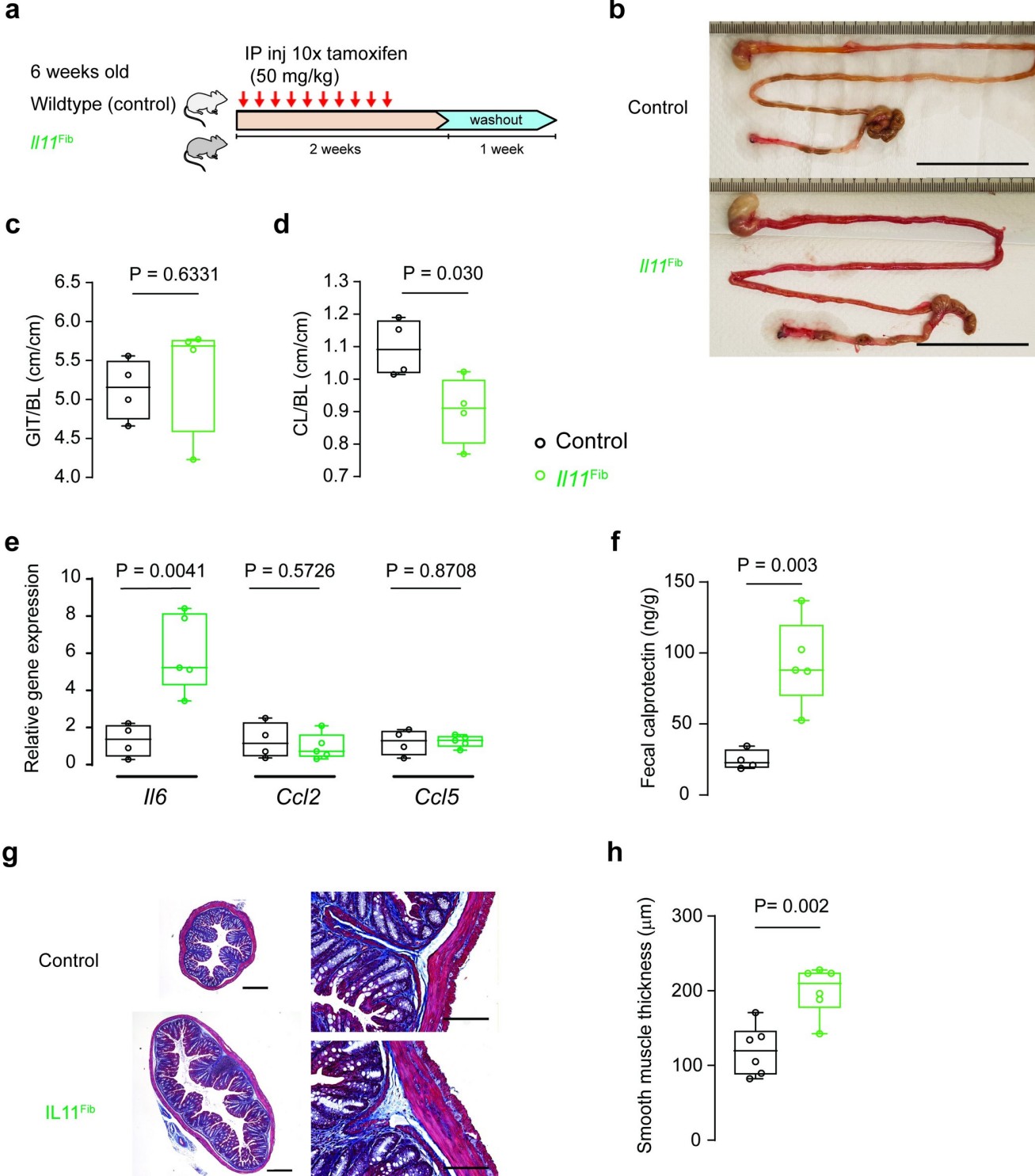

**Fig 8. Mice with fibroblast-specific *Il11* expression develop inflammatory bowel disease. (a)** Schematic diagram demonstrating tamoxifen (tam) injection procedure in 6-week-old *Il11*[Fib] and wildtype (control) littermates. **(b)** Excised gastrointestinal tract of representative *Il11*[Fib] mice at day 21 compared to controls. Scale bar represents 5 cm. **(c)** Indexed GIT length in reference to body length (BL) was unchanged in *Il11*[Fib] mice but **(d)** indexed colon length was markedly reduced as compared to controls (*n* = 4 per group). **(e)** Expression of inflammatory genes (*Il6*, *Ccl2* and *Ccl5*) in the colon tissue of *Il11*[Fib] mice as compared to controls (*n* = 4–5 per group). **(f)** Fecal calprotectin in stool samples collected from *Il11*[Fib] and control mice (*n* = 4–5 per group) as assessed by ELISA. **(g)** Representative cross-sections of the colon of *Il11*[Fib] and control mice stained with Masson's Trichrome (left) and at 200X magnification (right)

(*n* = 6 biological replicates). Scale bars indicate 500 μm and 200 μm respectively. **(h)** Thickness of the smooth muscle layer (muscularis propria) in tam-treated *Il11*[Fib] mice compared to controls (*n* = 6 per group). All comparisons were conducted in 21 days post-tam initiation in control (black) and *Il11*[Fib] (green) mice. Statistical analyses by two-tailed unpaired *t*-test; data expressed as median ± IQR, whiskers represent the minimum and maximum values.

colon [11, 40, 41]. Surprisingly, expression of *Il11* in SMCs was sufficient to induce severe colonic inflammation and rectal prolapse within 3 days, which was followed by early mortality in *Il11*[SMC] animals. We also documented increased colonic muscle thickness, which is a characteristic of the dextran sulphate sodium-induced colitis model [42]. In humans, histological features of clinical colitis include architectural distortion, shortening and size variation of crypts, immune cell infiltration, and granuloma formation [43]. Occurrences of architectural distortion of the glands and crypts in these mice were rare but present, although this may be reflective of the very short duration of IL11 expression. In contrast, these mice demonstrated thicker muscularis mucosa, increased immune cell infiltration, increased pro-inflammatory markers LAMP2 and LGALS3 in epithelial cells and the stroma and increased fibrosis in the mucosa sharing close similarities to intestinal fibrosis as observed in patients with ulcerative colitis [44]. Interestingly, IL11 expression in smooth muscle cells demonstrated signs of neuro-inflammation in the myenteric plexus, which has been observed in inflammatory bowel disease [45].

We explored further the IL11 effect in the bowel using an additional model that expresses mouse *Il11* in a second stromal cell type: the fibroblast. This complementary model also develops severe diarrhea and inflammation of the small intestines and colon, reinforcing the data generated in the *Il11*[SMC] mice. In this model, the colon becomes distended with thicker muscularis mucosa, suggesting that IL11 secreted from fibroblasts acts in a paracrine fashion to cause smooth muscle hypertrophy. A lack of grossly detectable intestinal fibrosis in the colon in this model, which is very different to findings in the heart, kidney and lung [8,9], may reflect differing cellular composition of fibroblasts and smooth muscle cells in the intestinal wall, where smooth muscle cells appear to play a larger role. This would be consistent with the suggestion that smooth muscle hyperplasia and hypertrophy contributes mostly to the fibrostenosis and inflammation in IBD [46] and underlies the colonic contractile dysfunction [47].

Taken together these data show that *Il11* expression in stromal cells is sufficient to cause an IBD phenotype and challenges the earlier data, based on the use of recombinant human IL11 in the mouse, that IL11 is protective in the bowel. Considered along with patient studies that show IL11 to be highly upregulated in the colonic mucosa of patients with ulcerative colitis or Crohn's disease and that IL11 predicts treatment failure [32–34], our results highlight IL11 as a promising therapeutic target for IBD, particularly in the context of anti-TNF therapy resistance.

## Supporting information

**S1 Table. Genotyping primers.**
(DOCX)

**S2 Table. RT-qPCR primers.**
(DOCX)

**S1 Fig. Comparison of tamoxifen-treated *Il11*[SMC] and *Cre*[SMC] mice. (a)** Schematic diagram demonstrating the tamoxifen (tam) injection procedure in 6-week-old *Il11*[SMC] and *Cre*[SMC] mice. **(b)** Survival curve of tam-treated *Il11*[SMC] (*n* = 35) compared to *Cre*[SMC] mice (*n* = 27) mice from 1st injection starting at 6 weeks of age. Survival curves were compared with the log-rank Mantel-Cox test. **(c)** Representative images of the *Cre*[SMC] and *Il11*[SMC] mice before (d0) and up to 14 days (d14) post-tam initiation (left). Note the presence of pale and loose stools in

*Il11*<sup>SMC</sup> mice (right). The presence of rectal prolapse is indicated with white arrows. Tam-treated *Il11*<sup>SMC</sup> images presented here are different from Fig 1c. Images were not taken to scale. **(d)** Baseline body weight of 6-week-old *Cre*<sup>SMC</sup> and *Il11*<sup>SMC</sup> mice before induction ($n$ = 16 per group). Statistical analyses by two-tailed unpaired $t$-test; data expressed as median ± IQR, whiskers represent the minimum and maximum values. **(e)** Representative images of *Cre*<sup>SMC</sup> and *Il11*<sup>SMC</sup> mice at d14 post-Tam initiation. **(f)** Collated body weights and **(g)** body lengths of tam-treated *Cre*<sup>SMC</sup> and *Il11*<sup>SMC</sup> mice measured at d14 post-Tam initiation. ($n$ = 12–17 per group). Statistical analyses by two-tailed unpaired $t$-test; data expressed as median ± IQR, whiskers represent the minimum and maximum values.
(TIF)

**S2 Fig. Uncropped blots for PCR genotyping from *Cre*<sup>SMC</sup> and *Il11*<sup>SMC</sup> mice treated with either tamoxifen (Tam) or vehicle (Veh).** PCR products of DNA extracted from tail biopsies of 21-day-old mice by use of set primers for *Myh11-Cre* (left) and *Rosa26-Il11* (right) (primers as listed in S1 Table) and analyzed by agarose gel electrophoresis. Dashed boxes indicate cropped blots used in Fig 1c.
(TIF)

**S3 Fig. Immunohistochemistry with rat and rabbit IgG isotype controls as negative staining control in Fig 2h.** Smooth muscle and crypt staining with rat and rabbit IgG isotype controls demonstrate no positive staining in both veh- and tam-treated *Il11*-Tg colon. Scale bar represents 100 μm.
(TIF)

## Acknowledgments

The authors would like to acknowledge B.L. George, E. Khin, M. Wang, J. Tan for their technical expertise and support.

## Author Contributions

**Conceptualization:** Wei-Wen Lim, Benjamin Ng, Liping Su, Stuart Alexander Cook, Sebastian Schafer.

**Data curation:** Wei-Wen Lim, Benjamin Ng, Stuart Alexander Cook, Sebastian Schafer.

**Formal analysis:** Wei-Wen Lim, Benjamin Ng, Anissa Widjaja.

**Funding acquisition:** Stuart Alexander Cook, Sebastian Schafer.

**Investigation:** Wei-Wen Lim, Benjamin Ng, Anissa Widjaja, Chen Xie, Liping Su, Nicole Ko, Sze-Yun Lim, Xiu-Yi Kwek, Stella Lim.

**Methodology:** Wei-Wen Lim, Benjamin Ng.

**Project administration:** Wei-Wen Lim, Stuart Alexander Cook, Sebastian Schafer.

**Resources:** Benjamin Ng, Stuart Alexander Cook.

**Supervision:** Stuart Alexander Cook, Sebastian Schafer.

**Validation:** Wei-Wen Lim, Benjamin Ng, Liping Su, Stuart Alexander Cook, Sebastian Schafer.

**Visualization:** Wei-Wen Lim, Benjamin Ng, Anissa Widjaja, Stuart Alexander Cook, Sebastian Schafer.

**Writing – original draft:** Wei-Wen Lim, Benjamin Ng, Stuart Alexander Cook, Sebastian Schafer.

**Writing – review & editing:** Wei-Wen Lim, Benjamin Ng, Stuart Alexander Cook, Sebastian Schafer.

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
