## [Decision Letter · Decision Letter 0]

4 Nov 2019

PONE-D-19-24461

Interleukin 11 expression causes murine inflammatory bowel disease

PLOS ONE

Dear Dr Lim,

Thank you for submitting your manuscript to PLOS ONE. After careful consideration, we feel that it has merit but does not fully meet PLOS ONE’s publication criteria as it currently stands. Therefore, we invite you to submit a revised version of the manuscript that addresses the points raised during the review process.

We would appreciate receiving your revised manuscript by Dec 19 2019 11:59PM. To enhance the reproducibility of your results, we recommend that if applicable you deposit your laboratory protocols in protocols.io, where a protocol can be assigned its own identifier (DOI) such that it can be cited independently in the future. For instructions see: http://journals.plos.org/plosone/s/submission-guidelines#loc-laboratory-protocols

We look forward to receiving your revised manuscript.

Kind regards,

Cristiano Pagnini, M.D, PH.D.

Academic Editor

PLOS ONE

Journal Requirements:

2. At this time, we request that you  please report additional details in your Methods section regarding animal care, as per our editorial guidelines. Please describe any steps taken to minimize animal suffering and distress, such as by administering anaesthesia, during the intraperitoneal injections of tamoxifen". Thank you for your attention to this request

Additional Editor Comments (if provided):

The paper is interesting but has consistent flaws, as highlighted in particular by reviewer#1. If the authors are able to address the criticisms and improve the paper, it can be reconsidered for publication.

Reviewers' comments:

Reviewer's Responses to Questions

**Comments to the Author**

1. Is the manuscript technically sound, and do the data support the conclusions?

Reviewer #1: No

Reviewer #2: Yes

2. Has the statistical analysis been performed appropriately and rigorously? 

Reviewer #1: Yes

Reviewer #2: Yes

3. Have the authors made all data underlying the findings in their manuscript fully available?

Reviewer #1: No

Reviewer #2: Yes

4. Is the manuscript presented in an intelligible fashion and written in standard English?

Reviewer #1: Yes

Reviewer #2: Yes

5. Review Comments to the Author

Reviewer #1: In my opinion this manuscript present major multiple concerns. Of note, while the most striking feature of this mouse model is supposed to be the development of intestinal inflammation resembling inflammatory bowel disease (IBD) (as highlighted in the title), the main feature seems to be involving global and multi-organ abnormalities leading to 37% of surviving mice to day 14. The authors try to describe the development of an IBD-like phenotype with just few supporting data that actually reduce my confidence in the overall study.

1)A great deal rests on the characterization of intestinal inflammation. Fecal calprotectin should not be used as the only parameter to properly assess intestinal inflammation. A more detailed assessment of intestinal inflammation should be performed by an expert GI pathologist, who should be able to provide details about location of the inflammation (small or large intestine), active inflammation, chronic inflammation, transmural inflammation, percentage of ulceration etc.

2) Fibrosis is typically a long process. How do the authors explain the high severity of fibrosis in the different organs following just 1 or 2 weeks of tamoxifen injection?

3)In figure 2d, why is the color of the of stained section so different? It looks like a different method for staining was used.

4)In figure 3 the western blots are not convincing. The author should show in the main figure the whole gel but not just sections of it. P-STAT3 has been described in the results to be unchanged in heart, lung and liver. Looking at the blots, it looks like that also in the skin there are not changes. Quantification of the bands and statistical analysis should have been also reported in main figure.

5)How much IL-11 different tissues make? Is there a way to measure IL-11 protein production from tissue explants? Can the author stain (i.e. by IHC) IL-11 and correlate with levels of intestinal inflammation and fibrosis?

6)It is said in multiple places that IL11 secretion from smooth muscle cells drive fibrogenic gene expression. How IL-11 secretion was measured?

7)Similarly, which cells in the intestine express the receptor for IL-11 and is there a hint about the function?

8)Again, a more detailed assessment of intestinal inflammation should be performed by an expert GI pathologist for the second model described.

9)The main effect of IL11 expression seems to be induction of fibrosis across organs. This is not recapitulated in the second model of IL11 expression in fibroblasts. Is the level of calprotectin enough to claim that the second model recapitulates the colonic inflammatory phenotype seen in IL11smc mice? This is a major concern.

Reviewer #2: With this manuscript Authors examined the effects of SMC-specific, conditional expression of murine IL11 in a transgenic mouse (Il11SMC). They show that IL11 secretion from the stromal niche is sufficient to drive an IBD-like inflammation in mice. Overall, the manuscript is very well written and the data and overall message are convincing and an important advance in the field. The methods are robust, and the results support the conclusions.

Authors should however better discuss the histologic and mucosal differences between the model presented and IBD in order to weight the real impact of this model on the knowledge of IBD pathogenesis.

6. PLOS authors have the option to publish the peer review history of their article (what does this mean?). If published, this will include your full peer review and any attached files.

Reviewer #1: No

Reviewer #2: No

---

## [Author Response · Author response to Decision Letter 0]

16 Nov 2019

Dear Cristiano Pagnini,

Thank you for sending our paper to review, we are grateful for the opportunity to improve our manuscript further and trust that the updated version addresses your concerns.

The manuscript now follows PLOS ONE’s style requirements and we report additional details in your Methods section regarding animal care. Original uncropped and unadjusted blot/gel image data were in S2 Fig (which has been moved into the main figure Fig 3 following reviewer #1 comments).

We were extremely encouraged by the comments of Reviewer 2 who stated “Overall, the manuscript is very well written and the data and overall message are convincing and an important advance in the field. The methods are robust, and the results support the conclusions”. This is in keeping with some reviews of our recent work at other journals. We were, therefore, most surprised by the comments of Reviewer 1 that are very much at odds with those of Reviewer 2, indeed at the opposite end of the spectrum. We believe Reviewer 1’s comments are misplaced and based on some erroneous interpretation of the data. 

For example, reviewer 1 ignores RNA-based inflammation markers that we show throughout the manuscript. The reviewer also disregards the clinically important disease endpoints of IBD such as florid diarrhoea, rectal prolapse, and bleeding and asks instead for more detail of intermediate phenotypes, which do not predict disease severity or patient outcomes. Taken together with the well-established marker for intestinal inflammation calprotectin, our data show that the genetic models develop an IBD phenotype. IL11 is highly upregulated in human IBD patients - notably in stromal cells in the bowel wall - and its expression predicts treatment failure (we cite three studies showing this) and thus our data showing that IL11 causes the IBD with clinically relevant disease phenotypes will be of great interest: it establishes IL11 as an IBD disease gene for the first time. 

Despite these concerns, we have addressed the points raised by the reviewers. In particular, we performed a number of new experiments and provide substantial additional data and analyses to address reviewer 1’s comments. The manuscript has been revised accordingly and is further improved. We also adjusted the title to specifically address a concern of Reviewer 1. We point out reviewer 2 asked only for a change to the discussion, which we provide.

We strongly believe that this manuscript will be of broad interest to the readership of the journal and presents novel and clinically relevant results that may influence therapy.

Please find below our detailed responses and we look forward to hearing from you soon.

Sincerely,

Sebastian Schäfer and Stuart Cook

 

Point-by-point response the Reviewers

Reviewer #1: In my opinion this manuscript present major multiple concerns. Of note, while the most striking feature of this mouse model is supposed to be the development of intestinal inflammation resembling inflammatory bowel disease (IBD) (as highlighted in the title), the main feature seems to be involving global and multi-organ abnormalities leading to 37% of surviving mice to day 14. The authors try to describe the development of an IBD-like phenotype with just few supporting data that actually reduce my confidence in the overall study.

While we focused our studies on the most prominent phenotype in transgenic mice - the IBD phenotype - we studied multiple organs and reported data across these organs. As stated in the abstract “The bowel of Il11SMC mice was inflamed, fibrotic and had a thickened wall, …. In other organs, including heart, lung, liver, kidney and skin there was a phenotypic spectrum of fibro-inflammation.” As such, this manuscript is not only about the bowel and IL11 is thought to be important in a range of fibro-inflammatory diseases. However, the bowel phenotype was the most prominent and, see our next answer, the upregulation of IL11 in IBD patients makes this aspect of the animal model particularly exciting. We disagree with the reviewer that we presented “just few” supporting data for the IBD phenotype and now provide a large amount of additional layers of evidence, as requested by this reviewer (see below). To address the reviewers concerns regarding the overall message of the manuscript, we have adjusted the title to include reference to the cross-tissue inflammation while retaining the focus of the manuscript on the overt IBD phenotype. The revised title now reads “Transgenic interleukin 11 expression causes cross-tissue fibro-inflammation and an inflammatory bowel phenotype in mice”. We trust this addresses the reviewer’s concern on this matter. 

1) A great deal rests on the characterization of intestinal inflammation. Fecal calprotectin should not be used as the only parameter to properly assess intestinal inflammation. A more detailed assessment of intestinal inflammation should be performed by an expert GI pathologist, who should be able to provide details about location of the inflammation (small or large intestine), active inflammation, chronic inflammation, transmural inflammation, percentage of ulceration etc.

The manuscript does not rely solely on the characterization of intestinal inflammation. It is an established fact that IL11 is the most upregulated cytokine in non-responding patients suffering from Crohn’s Disease or Ulcerative Colitis (see Arijs et al Gut 2009, Smillie et al. Cell 2019, Arijs et al. Infl. Bow. Dis. 2010). Thus, by definition, IL11 is a gene relevant in a range of IBDs. This manuscript addresses a critical gap in the literature, where no gain-of-function studies exist that examine the effect of species-matched mouse IL11 on the murine gut. We overexpress mouse IL11 in the mouse in two cell-types important for IBD: Fibroblasts and SMCs. Hence, this model should give important insights into the pathogenesis of human disease related to the disease gene IL11. 

We point out that this is a rapid model that develops in weeks and not a chronic inflammation model. Inflammation is present throughout the bowel in both models, as shown in Fig 2b and new Fig 8b, however we have focused on the colon phenotype due to overt characteristics of rectal prolapse suggesting the lower bowels to be highly inflamed. Additionally, while there is no obvious ulceration in our models, we have however noticed significant leukocyte aggregation structures and inflammation in some sections - see new data Fig 2h and Fig 3- and mice have died with blood in the intestinal cavity that suggests acute ulceration and bleeding that results in death. 

In addition, we did not rely solely on fecal calprotectin to assess intestinal inflammation but also profiled other well established pro-inflammatory markers such as Il-6, Ccl2, and Ccl5 on the RNA level (see Fig. 6), which are all significantly elevated in the bowel of Il11SMC mice. Gross histology also clearly shows red and swollen intestines that are clearly indicative of a prominent inflammatory response in this organ. Most importantly, IBD is characterized by a number of pathologies in addition to inflammation. The Il11SMC model presented here also presents with acute large and small intestine disease (as shown in Fig 2b) along with florid diarrhea, rectal prolapse, and bleeding. This is accompanied by prominent fibrosis. 

In the revised manuscript, we have followed up on the reviewer’s comments on other markers of inflammation and immunohistochemically stained for IL11, together with CD45 (leukocytes), LAMP-2, and LGALS3, markers that are commonly present in activated epithelial cells and immune cells. These, notably LGALS3, are elevated in the Il11SMC colon and there is widespread activation within Payer’s patches upon IL11 expression in SMCs (new Fig 2h and new Fig 3).

2) Fibrosis is typically a long process. How do the authors explain the high severity of fibrosis in the different organs following just 1 or 2 weeks of tamoxifen injection?

IL11 is an extremely potent pro-fibrotic factor. We have shown previously that IL11 can induce fibrosis in the cardiovascular system (Schafer et al. Nature 2017), lung (Ng et al. STM 2019) and liver (Widjaja et al. Gastroenterology 2019) within just weeks. Transgenic overexpressing of IL11 in fibroblasts or SMCs was hypothesized to be highly fibrogenic and the results shown in the manuscript are in line with previous observations. The data also show that IL11 expression in the stromal compartment is pro-inflammatory, which is a novel finding. 

3)In figure 2d, why is the color of the of stained section so different? It looks like a different method for staining was used.

We thank the reviewer for raising this point. Identical methods for staining were used, but sections were done in batches which resulted in slight color differences. Images have been replaced with representative images from identical batches.

4)In figure 3 the western blots are not convincing. The author should show in the main figure the whole gel but not just sections of it. P-STAT3 has been described in the results to be unchanged in heart, lung and liver. Looking at the blots, it looks like that also in the skin there are not changes. Quantification of the bands and statistical analysis should have been also reported in main figure.

We thank the reviewer for raising this point and agree with the suggestion that will improve the manuscript. We have now moved the uncropped blots from Fig S2 to Fig 4 along with the densitometric analyses. Quantification and statistical analysis are now also shown in the main part of the manuscript. Dermal stat activation is weak, but densitometry and statistical analyses indicate that P-STAT3 is slightly elevated in the skin. Thus we are hesitant to state there are no changes.

5)How much IL-11 different tissues make? Is there a way to measure IL-11 protein production from tissue explants? Can the author stain (i.e. by IHC) IL-11 and correlate with levels of intestinal inflammation and fibrosis?

We showed IL11 protein upregulation across tissues with quantification and statistics (previous Fig 3, now Fig 4). However, the actual level of IL11 production and its effects on the tissue level is highly dependent on the cellular composition of both the IL11 source (SMCs/fibroblasts) and the target cells (SMCs/fibroblasts/epithelium etc). As there are many possible cellular targets and the exact tissue composition is unknown, we do not believe that a simple correlation will provide any useful insights and point to the clinically relevant endpoints as the most important and defining phenotypes. It is the case that following this first definitive report of IL11 as an IBD gene that further and more detailed studies of cell-type effects will need to be studied using conditional gene knockouts and reporter strains to dissect inflammation-fibrosis cross-talk, which we are currently planning. We hope the reviewer can appreciate this fact.

6)It is said in multiple places that IL11 secretion from smooth muscle cells drive fibrogenic gene expression. How IL-11 secretion was measured?

IL11 is a secreted protein by its very nature as a cytokine via a protein motif that targets it to the extracellular space. The genetic models we utilize specifically induce IL11 protein expression in the target cells (either SMCs or fibroblasts). Western blots clearly showed upregulation of IL11 protein.

7)Similarly, which cells in the intestine express the receptor for IL-11 and is there a hint about the function?

The reviewer raises an interesting question and there is more than one answer. Smillie et al. Cell 2019 performed a single-cell study of human UC colon and found IL11RA expressed predominantly on fibroblasts and smooth muscle cells. The effect of IL11 on fibroblasts has been well documented by our group in a number of studies. More recently, we have found that IL-11 on fibroblasts is a potent driver of stromal inflammation: Ng B et al. Fibroblast-specific IL11 signaling is required for lung fibrosis and inflammation bioRxiv doi: https://doi.org/10.1101/801852

We suspect that this also plays a role in IBD, as Smillie et al. have also found an “Inflammatory Fibroblast” to be underlying anti-TNF resistance in IBD. However, this requires further study and is beyond the scope of this manuscript. We point out that IL11RA is also expressed on epithelial cells at high levels and it could well be that IL11 secreted from the stroma then impinges on the IECs but investigating this will again require further study.

8)Again, a more detailed assessment of intestinal inflammation should be performed by an expert GI pathologist for the second model described.

As with the first Il11SMC model, we show inflammation in the second Il11Fib model via calprotectin as well as elevated RNA levels of IL6. We now also show gross histology of this model, which clearly indicates a swollen and inflamed bowel that recapitulates the first model (revised Fig 7b). A significant reduction in colon length was also indicative of murine colitis, which was also shown. Again we would like to point out that the clinically meaningful endpoint of disease is ultimately more important than intermediate phenotypes that may or may not be present in animal models that variably recapitulates all the salient features of human disease. 

9)The main effect of IL11 expression seems to be induction of fibrosis across organs. This is not recapitulated in the second model of IL11 expression in fibroblasts. 

We cited a number of publications where we have previously shown that the second model (Il11Fib) develops severe fibrosis: Cardiovascular (Schafer et al. Nature 2017), pulmonary (Ng et al. STM 2019) and liver (Widjaja et al. Gastroenterology 2019) fibrosis. Perhaps the reviewer missed our citation of these papers that explain this model in some more detail. We point out that it is not expected that transgenic models with Il11 expression in SMC or fibroblasts will behave identically as these are - by definition - different cell types with different functions. Indeed SMC may be regarded as a more ‘inflammatory’ cell type as compared to fibroblasts, by some (Ng et al. Mediators Inflamm. 2003 and Chen et al. J Crohns Colitis 2017).

Is the level of calprotectin enough to claim that the second model recapitulates the colonic inflammatory phenotype seen in IL11smc mice? This is a major concern.

As we discuss in response to questions above, we do not base our observations solely on the elevation of calprotectin. We also do not want to argue, that the Il11Fib or the Il11SMC model are exact phenocopies. Indeed, they are different models and should be investigated as such. We have now amended the text to make this clearer and would like to thank the reviewer for pointing this out. We now present additional data on the second model (revised Fig 8).

Reviewer #2: With this manuscript Authors examined the effects of SMC-specific, conditional expression of murine IL11 in a transgenic mouse (Il11SMC). They show that IL11 secretion from the stromal niche is sufficient to drive an IBD-like inflammation in mice. Overall, the manuscript is very well written and the data and overall message are convincing and an important advance in the field. The methods are robust, and the results support the conclusions.

We thank the reviewer for his supportive comments.

Authors should however better discuss the histologic and mucosal differences between the model presented and IBD in order to weight the real impact of this model on the knowledge of IBD pathogenesis.

We now discuss the histologic differences between the Il11SMC and Il11Fib models and human IBD in greater detail in the manuscript. However, we would like to point out that IL11 upregulation in the colon has been repeatedly detected in IBD patients (see Arijs et al Gut 2009, Smillie et al. Cell 2019, Arijs et al. Infl. Bow. Dis. 2010). Thus, the mechanisms present in the IL11 transgenic animals are highly clinically relevant, even if they do not completely phenocopy human disease (not uncommon in rodent models).

---

## [Decision Letter · Decision Letter 1]

20 Dec 2019

Transgenic interleukin 11 expression causes cross-tissue fibro-inflammation and an inflammatory bowel phenotype in mice

PONE-D-19-24461R1

Dear Dr. Lim,

We are pleased to inform you that your manuscript has been judged scientifically suitable for publication and will be formally accepted for publication once it complies with all outstanding technical requirements.

With kind regards,

Cristiano Pagnini, M.D, PH.D.

Academic Editor

PLOS ONE

Additional Editor Comments (optional):

Reviewers' comments:

Reviewer's Responses to Questions

**Comments to the Author**

1. If the authors have adequately addressed your comments raised in a previous round of review and you feel that this manuscript is now acceptable for publication, you may indicate that here to bypass the “Comments to the Author” section, enter your conflict of interest statement in the “Confidential to Editor” section, and submit your "Accept" recommendation.

Reviewer #2: All comments have been addressed

2. Is the manuscript technically sound, and do the data support the conclusions?

Reviewer #2: Yes

3. Has the statistical analysis been performed appropriately and rigorously? 

Reviewer #2: Yes

4. Have the authors made all data underlying the findings in their manuscript fully available?

Reviewer #2: Yes

5. Is the manuscript presented in an intelligible fashion and written in standard English?

Reviewer #2: Yes

6. Review Comments to the Author

Reviewer #2: (No Response)

7. PLOS authors have the option to publish the peer review history of their article (what does this mean?). If published, this will include your full peer review and any attached files.

Reviewer #2: No

---

## [Editor Report · Acceptance letter]

26 Dec 2019

PONE-D-19-24461R1 

Transgenic interleukin 11 expression causes cross-tissue fibro-inflammation and an inflammatory bowel phenotype in mice 

Dear Dr. Lim:

I am pleased to inform you that your manuscript has been deemed suitable for publication in PLOS ONE. Congratulations! Your manuscript is now with our production department. 

With kind regards,

on behalf of

Dr. Cristiano Pagnini 

Academic Editor

PLOS ONE